# Redrawing Cities with Children and Adolescents: Development of a Framework and Opportunity Index for Wellbeing—The REDibuja Study Protocol

**DOI:** 10.3390/ijerph19095312

**Published:** 2022-04-27

**Authors:** Nicolas Aguilar-Farias, Francisca Roman Mella, Andrea Cortinez-O’Ryan, Jaime Carcamo-Oyarzun, Alvaro Cerda, Marcelo Toledo-Vargas, Sebastian Miranda-Marquez, Susana Cortes-Morales, Teresa Balboa-Castillo

**Affiliations:** 1Department of Physical Education, Sports and Recreation, Universidad de La Frontera, Temuco 4780000, Chile; andrea.cortinez@ufrontera.cl (A.C.-O.); jaime.carcamo@ufrontera.cl (J.C.-O.); marcelo.toledo@ufrontera.cl (M.T.-V.); sebastian.miranda@ufrontera.cl (S.M.-M.); susana.cortes@ufrontera.cl (S.C.-M.); 2UFRO Activate Research Group, Universidad de La Frontera, Temuco 4780000, Chile; 3Centro de Investigación en Epidemiología Cardiovascular y Nutricional (EPICYN), Universidad de La Frontera, Temuco 4780000, Chile; alvaro.cerda@ufrontera.cl (A.C.); teresa.balboa@ufrontera.cl (T.B.-C.); 4Department of Psychology, Universidad de La Frontera, Temuco 4780000, Chile; maria.roman@ufrontera.cl; 5Department of Basic Sciences, Universidad de La Frontera, Temuco 4780000, Chile; 6Vicerrectoría de Investigación y Postgrado, Universidad de La Frontera, Temuco 4780000, Chile; 7Department of Public Health, Universidad de La Frontera, Temuco 4780000, Chile

**Keywords:** wellbeing, quality of life, urban health, inequity, health in all policies

## Abstract

Global changes require urgent integration of health and wellbeing into all urban policies. Complex social and environmental factors define wellbeing outcomes and inequities present in cities. Additionally, political decisions are seldom thought and developed considering the needs and participation of children and adolescents. The REDibuja study aims to develop a multidimensional framework of wellbeing for children and adolescents and to validate an index of opportunities for better wellbeing for children and adolescents in the urban context of Temuco, Chile. This child-centered and cross-sectional study will involve mixed methodologies throughout the implementation of five work packages for two years (2022–2023): (1) development of a conceptual framework for child and adolescent wellbeing, (2) integration of available and public data, (3) studies in the local context, (4) data integration using geographic information systems, and (5) validation of the wellbeing opportunity index for children and adolescents. REDibuja will implement methodologies that until now are little used to facilitate political decisions in our regional context. This process and results could be transferred for assessment and decision-making in Latin America and low- and middle-income countries in other regions.

## 1. Introduction

The rapid increase in urbanization worldwide has various problematic impacts on the environment and urban communities. As a result, there are international calls for developing, planning, and implementing policies and strategies to improve the sustainability of cities as well as health and wellbeing in communities. Health and wellbeing are overlapping concepts, and the World Health Organization (WHO) defines health as “a state of complete physical, mental and social wellbeing and not merely the absence of disease or infirmity” [1]. In addition, there is no consensus on the definition of wellbeing, and the concept has been changing over time. Recently, during the coronavirus disease 2019 (COVID-19) pandemic era, wellbeing has been defined as a “state of positive feelings that can be measured subjectively and objectively, using a salutogenic approach” [2]. Moreover, modern research theories on wellbeing have not focused on addressing the childhood-derived approach. To make this possible, it is necessary to collect robust evidence to inform and move this agenda forward while promoting health and wellbeing equitably in this population [3]. These actions should be in line with the growing interest in understanding how the promotion of population health at the local and global level can contribute to achieving the Sustainable Development Goals (SDGs), considering its multidimensional benefits at the environmental, educational, economic, and social levels.

The need for greater integration of wellbeing into all urban policies, planning, and investments has attracted the attention of academics, regional, and global organizations [4]. The scientific literature shows different ways in which urban policies and health (including physical and mental health) interact [3]. Beyond individual lifestyles, complex social and environmental factors define health outcomes and inequities present in cities [5]. Many of these social and environmental determinants of health and wellbeing that reproduce inequities are the product of unequal budgetary conditions, local governance, urban planning, and the quantity and quality of interventions in cities [6].

In this context, children and adolescents are a group that depends to a great extent on the decisions, options, and opportunities to which they are exposed according to the context in which they are born and develop together with their families [7]. Socioeconomic factors and the environments where children and adolescents live, learn, and play (homes, neighborhoods, and schools), as well as the ways in which they move between these, affect children’s health and wellbeing. This is reflected in indicators such as nutritional status, mental health, cognitive and language development [6,8]. For example, residential segregation, a factor associated with racial, ethnic, and socioeconomic inequalities, explains not only health and wellbeing outcomes but also educational opportunities for children and adolescents [9]. Residential segregation limits equal access to available resources, affects the academic opportunities of children and adolescents, and reproduces processes of social exclusion affecting the most vulnerable groups, such as children and adolescents.

However, political decisions are seldom thought and developed considering the needs and participation of children and adolescents. The COVID-19 pandemic has demonstrated and exacerbated the negative impacts of not anticipating or considering the unintended effects that decisions at various policy levels have on children and adolescents’ health and wellbeing [10]. International experts have demanded urgent actions to counteract these effects on the integral development of children and adolescents [11].

Community development and interventions to improve neighborhood environments can help reduce inequities in health and wellbeing in children and adolescents [12,13,14,15]. However, more research is needed in Latin America to inform policy and decision-making [16]. Studies should not only consider common territorial attributes (e.g., neighborhood poverty, caregiver education) but also include information on the broader set of material and social opportunities provided by neighborhoods (access to squares, parks and green areas, safe spaces to play, playgrounds, nursery schools and housing quality, safety and risk perception by children and parents, etc.) that could be beneficial for the health and wellbeing of children and adolescents. They should also include not only standardized notions of wellbeing but also situated understandings based on the views of targeted groups such as children and their families in particular locations.

From an ecological perspective, children and adolescents’ development is the result of a continuous process of reciprocal exchange with the environment of which they are a part [17,18]. The wellbeing of children and adolescents is strongly influenced by the quality of relationships and resources in their family, school, and community environment [6,17]. In turn, family and school contexts are shaped by the economic, political, and social macro-system. Scholars and organizations have designed frameworks to address the multifactorial and multidimensional nature of children’s wellbeing and help in the design and implementation of actions [19,20]. For example, the Australian Research Alliance for Children and Youth (ARACY) has developed an evidence-based policy framework for promoting the wellbeing of children and adolescents. This implies simultaneously understanding their emotional (feeling loved and safe) and material needs, as well as attending to issues related to health, learning, participation, sense of identity, and culture [19].

Physical activity and play are essential activities for the development and health of children and adolescents. Family, material, and social environments have an important role in this regard [21,22]. For example, evidence indicates that a higher population density is associated with low levels of motor competence and a shorter time spent outdoors [23]. In the case of the Araucanía region, in the south of Chile, preliminary results support these trends, where schoolchildren from large cities have less motor skills than boys and girls from smaller cities; at the same time, children of low socioeconomic status have lower levels of physical activity, fewer hours of physical education at school, and lower levels of motor competence than boys and girls of high socioeconomic status [24,25]. However, these outcomes have not been analyzed in relation to residential segregation, a key factor shaping large cities in Chile [26] and other Latin American countries [27,28].

Following a similar trend, excess malnutrition, a cardiometabolic risk factor closely related to alterations in the regulation of glucose and lipid metabolism associated with the development of chronic conditions in adulthood, is also influenced by social and built environments [8,29]. Obesogenic environments and food deserts (areas with limited access to nutritious and affordable food) are closely associated with territorial inequities in relation to the prevalence of overweight and obesity, not only in children and adolescents but also in adults [30].

Unveiling the “unequal geography of opportunities” is especially relevant in highly unequal cities and societies such as Chile’s most populated cities [31,32]. In Chile, education and health show significant differences between private and public sectors in their resources, quality, and outcomes in the context of highly segregated territories [33,34,35,36]. Indices that integrate population data and wellbeing with geographical approaches in urban and rural settings are relevant sources of information for policy makers and researchers [37,38]. Internationally, diverse indices have been developed aiming at this goal [31,39,40,41], including the Urban Quality of Life Index in Chile, in which urban quality of life is measured in six dimensions, in ninety-nine municipalities, showing the provision of public and private goods and services from a socio-territorial perspective [42]. However, these indicators have historically focused on adult-based definitions of needs and wellbeing [43,44,45]. Aiming at bridging this gap, a network of researchers in the US developed the Child Opportunity Index (COI). The COI includes measures of (a) educational, (b) health and environmental, and (c) social and economic opportunities [31]. The application of the index concluded that in those metropolitan areas in the United States where there are more black and Hispanic children and adolescents, the indicators of opportunities are lower. For example, about 40% of black children and 32% of Hispanic children live in neighborhoods with very low opportunities, compared to only 9% of white children. The same study indicates that the metropolitan areas with the greatest inequities in opportunities were those that also showed high levels of residential segregation.

In Chile, due to the limited systematization of information and processing of data at the neighborhood level, decisions in public health and education are generally guided by coarser indicators elaborated at the urban or regional level. In turn, given the unequal distribution and quality of urban services capable of providing greater development opportunities for children and adolescents, such as schools, parks, transportation, and sports centers, among others, it is possible that it acts by increasing the gaps between already segregated strata. To our knowledge, there are no data on how these factors affect the health and wellbeing of children and adolescents in Chile or Latin America. For these reasons, the aims of the REDibuja study are to develop a multidimensional framework of wellbeing for children and adolescents and to validate an index of opportunities for better wellbeing for children and adolescents in the urban context of Temuco, Chile. Our study is inspired by previous initiatives with similar goals conducted in different places of the world, such as the COI [31], which has successfully helped in sparking conversations to address unequal opportunities by integrating data from the COI into strategic planning and resource allocation by local and state governments. However, we aim to construct this index through participatory means, so that children and adolescents’ views are included throughout the process. Therefore, the selection of indicators will be based on previous empirical and conceptual discussions around this but also on the experiences, knowledge, and perceptions of children and adolescents inhabitants of Temuco, as we further discuss in the next sections.

## 2. Materials and Methods

This cross-sectional study will involve mixed methodologies throughout the implementation of five work packages for two years (2022–2023). The REDibuja study will be conducted in Temuco, Chile, a city with 282.415 inhabitants in the south of the country [26], located about 620 km south of the Chilean capital of Santiago. Temuco is the regional capital of the poorest region in Chile [46].

The study will be child-centered but will include a diversity of participants in its different stages: children, adolescents, caregivers, community leaders, and professionals linked to educational or health services, and policy makers and implementers. For recruitment and analysis, children and adolescents will be categorized into the following age groups: 2 to 4 years old, 5 to 9 years old, 10 to 13 years old, 14 to 17 years old.

The mixed-methods design will combine surveys, ethnographic and participative approaches, and geographic information systems. The study has the approval of the Ethics Scientific Committee at Universidad de La Frontera, Chile (Act of approval: 142-2021, 3 November 2021).

### 2.1. The REDibuja Concept

The REDibuja concept is inspired by the relatively simple but meaningful action of drawing (*dibujar* in Spanish) and the sense of connection and impact of networks (*red* in Spanish) in people’s lives. Drawing is one step further than imagining or just thinking: drawing implies making visible and evidencing the feelings, experiences, perceptions, and perspectives implied in what constitutes a better place to live for children and adolescents as well as adults, who will be also invited to “redraw” a better city. The name of the project has been created before the fieldwork and participatory stages of the study begin; therefore, children have not been involved in choosing this name. However, we hope to engage participant children and adolescents in discussing what redrawing their city means, incorporating diverse ways of making sense (or not) of this name.

In addition to this, the notion of redrawing the city points towards the aim of dialoguing with urban and social policy makers, informing decision-making processes from the perspective of children’s and adolescents’ wellbeing in the city. For this purpose, we will capitalize on the qualitative methods of this study. Participatory drawings will allow children to depict concepts, emotions, and information in an empowering and personally relevant manner [47].

The findings of the participatory process, along with the index itself, will be finally presented to the council to become inputs to generate novel protocols and improve local policies with a focus on childhood. In this sense, we are working closely with Temuco’s council from the beginning of the project. This partnership will imply working collaboratively in planning and development of activities in the city, technical assistance/training for professionals, coordination of scientific, social, and cultural events. The results will be shared in different instances with key actors of public and private sectors, such as public and private schools, housing companies and organizations, urban planners, transport, health and education departments, families, and health centers. The study logo is shown in Figure 1.

### 2.2. Work Packages

The REDibuja study comprises five work packages that will be implemented following the steps and timeline shown in Figure 2. The description of each work package is provided below.

#### 2.2.1. Work Package 1 (WP1): Development of a Conceptual Framework for Child and Adolescent Wellbeing

Objective: to develop a conceptual framework of child and adolescent wellbeing implemented through a mixed approach that involves a review of the literature and community participatory research (Figure 3).

Design: The development of WP1 will be conducted in 6 stages based on the double diamond design model [34]: (1) Systematic review of the literature and summary of findings by the research team to develop the wellbeing framework proposals, (2) community participatory process to review the initial wellbeing framework proposals, including children’s and adolescents’ participation through interviews and focus groups, (3) integration of perspectives provided by the community, (4) presentation of the preliminary wellbeing framework, (5) revision and refinement of the conceptual framework by and with the community, and (6) presentation of the final framework.

Temuco is administratively divided into seventeen census districts. From these, eight will be selected based on aspects such as vulnerability, centrality, and population density, for the purposes of the participatory component of WP1. We aim to have a selection that represents the socioeconomic diversity of the city. In each selected district, and with the support of the Council, one school of each educational level (early years and primary and secondary education) will be recruited. Additionally, for each educational level in each school, we expect to recruit 10 children to participate in this stage of the study. This means that there will be approximately thirty children and adolescents participating in each census district and therefore an estimated 240 children and adolescents participants.

In addition to this, in each district, we will aim to count the participation of community members and key stakeholders: parents, local organizations representatives, teachers, school staff, and local authorities.

Methodology: The participation of children, adolescents, and other actors will be key in this process. To facilitate their participation, an array of qualitative and participative techniques will be used with different actors and according to the different educational levels of participant children and adolescents. Techniques will include: individual and group interviews (with a limited number of older children and adolescents, key stakeholders, and community members), participant observation with children, adolescents, and their families, creative techniques such as drawing and games (especially but not exclusively with younger children), and focus groups (with older children and adolescents). In addition to this, participants will have the opportunity to use the app Stanford Healthy Neighborhood Discovery Tool [48,49] both individually and as part of collective walks around schools and neighborhoods, so that they can register their perceptions of the places they inhabit. The participants will be asked to collect data through photos, audio, and/or text narratives and rating environmental attributes that may facilitate or hinder wellbeing. After data collection with the app, the research team and the participants will interact through meetings to discuss, analyze, and interpret their data. The main purpose will be identifying and building consensus about the most relevant environmental aspects that may favor or affect wellbeing in their living areas.

Ideally, this will be a face-to-face participatory process. However, remote or hybrid means may be implemented, depending on COVID-19 public health conditions and local regulations, and participants’ preferences and needs. This will involve the consideration of participants’ accessibility to remote and virtual means, in order to counteract the risk of exclusion of vulnerable groups of people from remote/virtual participatory research [50]. If this is the case, the Temuco Municipality will support us with technical and contact issues.

The aim of these techniques will be to gather information about the living conditions of participants in each census district, to present them the main findings of the systematic literature review in terms of models of child and adolescent wellbeing, and to generate opportunities for reflecting and discussing around the notion of wellbeing and appropriate indicators for constructing a situated index.

#### 2.2.2. Work Package 2 (WP2): Integration of Available and Public Data

Objective: to review databases available at the national, regional, and city levels and integrate them into the framework developed in WP1.

Design: data that could be integrated into the children’s and adolescents’ wellbeing index may include availability of educational establishments, index of school vulnerability, scores in educational outcomes assessment of the Agency for Quality of Education, educational level of adults, type of housing, overcrowding, accessibility—i.e., proximity to health centers and other relevant places, availability of public services, access and type of neighborhood stores, green areas, parks and squares, and so on, and adult employment rate.

Methodology: The data will be collected from repositories available in different ministries and through law 20.285 (access to public information) in Chile. In addition, information will be collected on variables related to the state of health and wellbeing of children and adolescents such as physical inactivity, sedentary behavior, nutritional status, cardiometabolic risk factors, and socio-emotional wellbeing, among others. The information will be harmonized in a database with the aim of integrating data at census districts or neighborhood levels. Children and adolescents will not be actively involved in this stage. However, the search of relevant data will be based on the participants’ views and discussion in WP1.

#### 2.2.3. Work Package 3 (WP3): Studies in the Local Context

Objective: To gather and generate new data to fill in possible information gaps detected according to the framework developed in WP1 and the data reviewed and integrated in WP2.

Design: There are previous and ongoing studies led by members of the research team that will contribute to developing WP3 in topics such as physical activity, motor competence, obesity, mental health, and substance consumption. In addition to this, educational establishments in eight districts across the city will be the focal points for conducting studies with children and adolescents. These studies will aim to create situated knowledge about specific themes that emerge as key in the constructions of the participatory children and adolescents’ wellbeing index. They may focus on particular territories within the city, specific kinds of spaces, problems, or themes. The exact character, aim, size, and techniques of these studies will be defined based on the needs detected in previous stages. However, the general methodological approach is outlined next.

Methodology: small-scale qualitative and participatory studies will be conducted from a mainly ethnographic approach. This means that researchers will work closely with participants through diverse techniques. These will vary depending on the specific study objectives, age and preferences of participants, and restrictions related to the COVID-19 pandemic. In general, techniques will include (but will not be limited to): use of the app Stanford Healthy Neighborhood Discovery Tool (see WP1) in individual and group modes, semi-structured interviews with children, adolescents, and other actors such as parents, teachers, school staff and local authorities, focus groups with students and other participants, participant observation, and games. Additionally, surveys will be designed and applied according to the emerging domains of wellbeing developed in WP1. The selection of questions for the surveys will be done through rapid reviews of the literature to find the most appropriate tool for the local context. An estimated sample of 1274 participants will be recruited considering a cluster sampling with a design effect of 1.3 [51], 80% rate of response, 3.5% of absolute precision, and 95% of confidence level.

As with WP1, this process will be preferentially face-to-face. If this is not possible due to the COVID-19 restrictions, remote methodologies will be implemented.

Data will be managed using REDCap (Research Electronic Data Capture) [52]. This platform allows offline data collection, facilitating the process when connectivity is limited. REDCap can be used with multiple devices (e.g., tablets, laptops, or mobile phones) and operative systems, providing flexibility and inclusion when collecting data.

#### 2.2.4. Work Package 4 (WP4): Data Integration Using Geographic Information Systems

Objective: To integrate data from WP2 and WP3 using geographic information systems (GIS) and to identify attributes of the environment that may influence children’s and adolescents’ wellbeing.

Design: Identified and collected data from WP2 and WP3 will be analyzed at the territorial level with an emphasis on identifying residential segregation and other forms of “unequal geographies of opportunity” in relation to children and adolescents’ wellbeing [31,32]. Depending on the territorial level (neighborhood, census area, sector, district, municipality), different layers of information will emerge. The summarized data will be open to the community on a web platform. No information will be published at the individual or residence level.

Methodology: Apart from the integration of the data collected in WP2 and WP3, environmental attributes that may promote or affect children and adolescents’ wellbeing will be identified through a rapid review. The findings of this review will be then validated using similar participatory approaches as shown in WP1 and WP3. The participatory activities will be conducted in contrasting areas of the city according to different socioeconomic statuses. The data collected from participants will be analyzed and compared with those derived from the rapid reviews using a thematic analysis approach [53] to select the attributes that should be included in the model (WP5).

#### 2.2.5. Work Package 5 (WP5): Validation of the Wellbeing Opportunity Index for Children and Adolescents

Objective: To validate the wellbeing opportunity index for children and adolescents in Temuco using data gathered through the previous work packages.

Design: Based on the framework developed in WP1, data generated through the other packages (WP2, WP3, and WP4), will be analyzed in relation to the resulting domains.

Methodology: An index or composite indicator can measure multidimensional phenomenon-compiling sub-indicators into a single index based on an underlying model [54]. First, for each domain of wellbeing opportunities (e.g., educational), the most appropriate sub-indicator(s) will be selected according to the conceptual framework built on WP1. Once the sub-indicators have been selected, we will follow several steps to build the index [54]. After assessing the quality of available data, we will examine the relationship between the sub-indicators using multivariate methods such as principal component analysis and factor analysis. Then, normalization should be conducted to make variables comparable. To analyze the robustness and sensitivity of the composite indicator, we will conduct sensitivity analysis to test whether changes in the sub-indicators and/or weighting may affect the results. Lastly, we will choose a visualization tool to facilitate the understanding of the composite indicator results. While WP5 does not entail the active participation of children and adolescents, the results will be shared and discussed with participants of previous stages. This will contribute to the final process of adjusting and validating the index.

### 2.3. Ethical Issues

As previously mentioned, the study has the approval of the Ethics Scientific Committee at Universidad de La Frontera, Chile (Act of approval: 142-2021, 3 November 2021).

Children and adolescents will be recruited through educational centers requiring the authorization of council authorities and educational centers’ principals. After obtaining the written authorization, educational centers will facilitate the promotion and recruitment process. In line with local ethical requirements, the qualitative strand of the study will work with an informed consent that will be signed by the parents or guardians of participant children and adolescents, and an informed assent that will be talked through with children and adolescents who decide to participate in the study. Both the consent of parents and assent of children will be required for them to be able to participate. In addition to this, researchers working in the field will be permanently available for questions or issues that may emerge and attentive to participants’ ongoing willingness to participate in the study. Additionally, key informants from the community will be part of the study. Caregivers, community leaders, professionals linked to educational or health services, and policymakers and implementers will sign an informed consent. All participants will be able to leave the study at any point if they desire to do so.

The study does not imply any foreseeable risks for participants. There will not be monetary payments or material compensations for participants, and their participation will be absolutely voluntary. However, the research team will organize dissemination events in which children and adolescents who have taken part in the research will be able to participate either publicly or as part of the organizing team. The research team will also organize—in partnership with council and university—workshops that may be of interest for the participants, according to their interests. The aim of the study is to improve the quality of life of children and adolescents in Temuco, and therefore this is a long-term benefit of taking part in the study.

## 3. Discussion

This manuscript summarizes the protocol of the REDibuja study. This mixed-method study will contribute to research by developing a multidimensional wellbeing framework for children and adolescents through participatory means in Temuco, Chile. This will highlight the opportunities and barriers found in this city for children and adolescents’ wellbeing. In doing so, this study will also help decision-makers by identifying wellbeing inequalities through the development and validation of an index of opportunities of wellbeing for children and adolescents in the urban context.

The study aims at bridging a gap, as available measures of health and wellbeing in this geographical area are based on adult-centric metrics (e.g., family income), presenting an immense challenge for building friendly and equitable cities for children and adolescents. This gap became even more critical in the context of the COVID-19 pandemic. It is in this context that the REDibuja study was born as an urgent response to the lack of inclusion and consideration of children and adolescents in the implementation of measures related to the COVID-19 pandemic. For example, in addition to the prolonged closure of schools and educational institutions, and the strict lockdown periods that affected children and adolescents in Chile [55], most recreational and green areas were restricted for several months. This seriously affected playing and active living practices in spaces other than homes. This may have largely affected children and adolescents living in low-income residential areas, in the context of overcrowded housing, unsafe streets, and lack of open/green areas [56]. These unprivileged conditions were exacerbated by the mobility restrictions imposed by governments at the beginning of the pandemic, so that people were not allowed to move towards other areas of the city. In the case of Chile, this affected children and adolescents living in urban areas in particular [57], as they did not have any permits for going out of their homes. This measure lasted up to six months in a row in some districts.

These circumstances highlight the need to better understand how urban environments and phenomena affect diverse communities, to inform future urban, health, and educational policymaking. It is key to consider the specific needs of diverse groups for counteracting collateral impacts on those that live in less friendly areas, less privileged conditions, or are part of minority groups such as children and adolescents. The use of this opportunity index for children and adolescents’ wellbeing will allow for better investment in solutions by considering the particularities and commonalities in urban areas. For example, playing areas may consider features that children find relevant for them, or maybe some environmental arrangements can be made to ensure a safe walk to school. Recently, in 2022, the local council has created the “Office of Childhood” whose mission is “to improve the conditions and quality of life of the child population, linking and relinking them with their families and communities and providing the necessary tools in the key stages of development” [58].

The main strength of the REDibuja study is the inclusion of children and adolescents in the development process of the wellbeing framework. We acknowledge the complexities of conducting a study of this kind, particularly when facing a global pandemic, and are aware that the aims of the study may be ambitious. However, we will work closely and in coordination with the Municipality of Temuco to facilitate the participation of key actors. Additionally, our methodological design considers remote options in the case that health safety and restrictions related to the COVID-19 pandemic require it. We will facilitate devices and we will coordinate connectivity options with schools and the council. Additionally, we will use Redcap [52], a platform that allows data collection without an internet connection. In the medium term, it would be feasible to provide a flexible data integration system capable of providing information according to the needs of each city or country in the region based on the information available at each site.

## 4. Conclusions

The REDibuja study will involve mixed methods to develop and validate a participatory framework and an index of opportunities for children and adolescents’ wellbeing in Temuco. This project will allow the implementation of methodologies that until now have been scarcely used to facilitate political decisions in our regional context. This process and results can be transferred for assessment and decision-making in Latin America and low- and middle-income countries in other regions. By redrawing a better place to live with the perspective of children and adolescents at the center, they will allow new opportunities to emerge not only for them but also for their entire communities.

## Figures and Tables

**Figure 1 ijerph-19-05312-f001:**
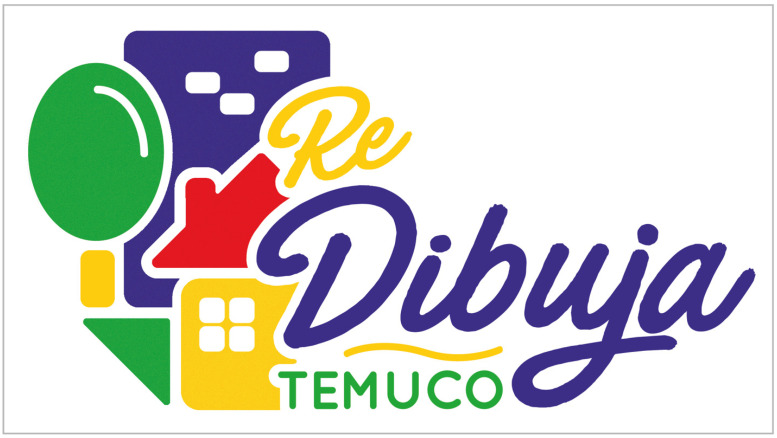
Logo of the REDibuja study.

**Figure 2 ijerph-19-05312-f002:**
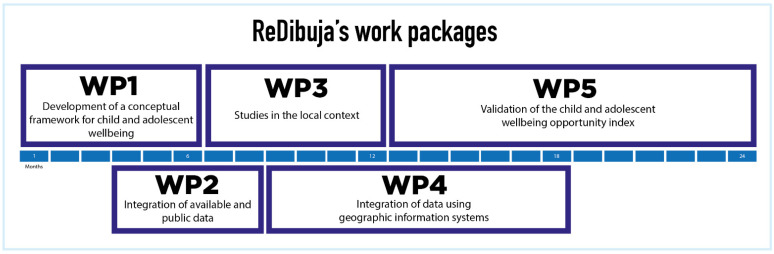
Work packages of the REDibuja study.

**Figure 3 ijerph-19-05312-f003:**
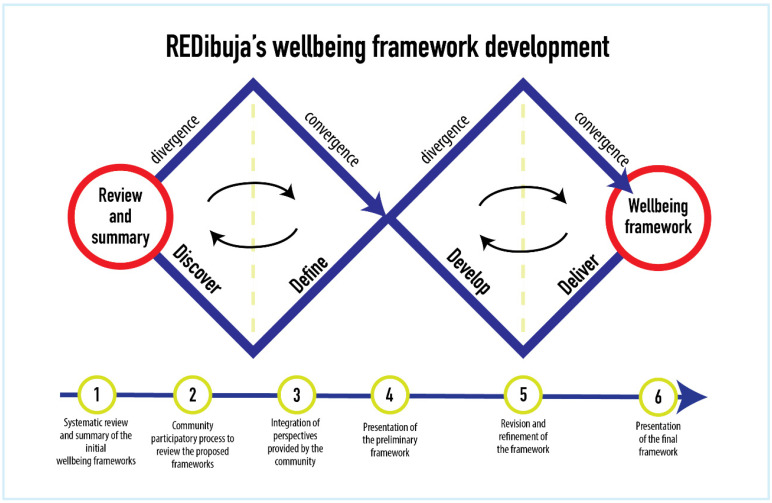
Double-diamond design model for the development of the REDibuja’s wellbeing framework.

## Data Availability

The anonymized data collected in this study will be available on specific requests from the corresponding author.

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
