# Peer review of "Redrawing Cities with Children and Adolescents: Development of a Framework and Opportunity Index for Wellbeing—The REDibuja Study Protocol"

_ijerph, 2022, doi:10.3390/ijerph19095312_

Round 1
Reviewer 1 Report
This is a very interesting paper which highlights extreme important issue on the development of a framework and opportunity index for wellbeing. The topic itself is relatively complex.
Nevertheless, I need to point out some main problems:
- This article doesn’t fit into any types of publications for the International Journal of Environmental Research and Public Health. It should be namely either Articles, Reviews or Case Reports. This submission is a Study Protocol, which mainly only documents the process and the design of a project.
- The main contents of this study protocol are a simple introduction of the project framework. The main theoretical background and the references to other studies and research are missing. The relevance to the other research is not yet established. I take an example. It is addressed that “a network of researchers from the US developed an index including measures of a) educational, b) health and environmental, and c) social and economic opportunities” (Line 121-123). In this circumstance the authors would like to refer to this design and run a simple project in Chile (line 130-142).
For me it is not clear if the opportunity index theoretically a reliable index is and the multidimensional framework of wellbeing possible is. I am also not sure if it is possible as there is a lack of information and systematic gathering or processing of data at the neighborhood level in Chile.
- My suggestion for the publication is to finalize first a literature review for this project and try to submit again.
Reviewer 2 Report
This is a very important topic and worthy of attention and research.
First, the article needs to be edited for grammar.
Second, the scope seems quite large for one article, but can work since it's just outlining the development and testing of the strategy. The work outlined in 2.2.2 is extensive and could take ages. I suggest eliminating the use of "among others" from the text, which appears a number of times, and really leaves the door quite wide open for any variety of other data sources, and is generally used to cover all other bases so as not to be criticized for forgetting something. However, it turns out to be largely meaningless. In a related note, it would be extremely helpful to see how this would look in an ideal timeline. This honestly sounds like it could be quite a long time, which is fine, but it should be noted.
In the Discussion, "This manuscript summarizes the protocol of the REDibuja study." But what it really does is provides a sketch, or high-level overview, of an anticipated study. There are no specific protocols outlined - particular sample groups or sizes, ages, etc. No number of surveys anticipated, number of times for focus groups, observations, and the like. For example, from 2.2.2: "...will be collected on variables related to the state of health and wellbeing of children and adolescents such as physical inactivity, sedentary behavior, nutritional status, risk factors for cardiometabolic, socio-emotional well-being, among others." Where are you getting this data? And what data exactly? This paper is too vague to truly offer a protocol.
I believe this paper can be worthwhile but should be reframed as a conceptualization of a possible framework. There is currently too much vagueness in the description - and this could be fine, but need to be appropriately framed.
Reviewer 3 Report
This is clearly written. The article presents the study protocol and makes the case for the study succinctly. The authors state that the main strength of the study is the inclusion of children and adolescents in the development process of the wellbeing framework. I suggest that this is an overstatement. I make the case that the study protocol should and could involve children and adolescents to a far greater extent. This will strengthen the scientific aspect of the study and it will ensure that the study meets its implicit aim of including the wellbeing of children and adolescents in decisions that affect their lives.
I run through the main components of the article below. I then offer some reflections.
Brief summary of the article and its contribution
This article calls for a rethink of the way that wellbeing for children and adolescents is framed. It sets out a protocol for a research project that will develop a conceptual framework for child and adolescent wellbeing; integrate public and available data; conduct local studies; integrate data in GIS; and finally, validate the data.
The title of the article ‘Redrawing cities with children and adolescents: development of a framework and opportunity index for wellbeing’ makes it clear that the focus is on the active involvement of children and adolescents in using their understanding of wellbeing to recreate the city.
The introductory section locates the study in the current context eg that of COVID-19 and the adverse effects that non-pharmaceutical interventions had on wellbeing for children and adolescents. The authors make the case that wellbeing needs to be integrated into urban policies, planning and investment decisions and how this is increasingly recognised in scientific literature. The authors refer to citizen science and the article by King et al (2021).
The authors establish how children and adolescents are dependent upon their environments and how political decisions rarely take account of their needs or seek to involve them.
Physical activity and play are given as examples of activities that are essential for development, but which can be constrained in densely populated areas. Impacts on motor skills and on levels of physical activity are shown. Links are also made between nutrition, the social and built environments, access to food and child and adult obesity. The authors establish how ethnicity and socioeconomic status are linked with many of the factors that frame opportunity, with black and Hispanic children and adolescents faring poorly.
The authors go on to describe the materials and methods that will make up the study. The study is described as participatory, and we are told what groups will be involved. These cover the different ages of children and adolescents and include caregivers, community leaders, professionals linked to education and health and policymakers as well as children and young people themselves.
The authors state they have ethical clearance.
The reader is shown the logo of the study. The authors explain the significance of the terms *red* and *dibujar* that make up the name of the study. It would be interesting to know whether children and adolescents were involved in choosing the name for the project or in designing the logo. Do the logo and project name hold the same significance for this age group?
The reader is taken through each of the work packages: WP1 to WP5. The approach to data management would seem appropriate eg develop a conceptual framework (WP1), see what data is already available (WP2), conduct local studies (WP3), put it all together in GIS (WP4) and validate (WP5).
The discussion re-states the case made in the introduction for the importance of the study. It notes that the main strength of the study is the inclusion of children and adolescents in the development process of the wellbeing framework.
Reviewer’s reflections
The article is well-written and easy to understand.
This is a study protocol. The structure of the article would appear to be appropriate.
There is no formal review of the literature. The authors summarise the literature by way of establishing the importance of the topic. A review of the literature will be one of the outcomes of WP1.
The article is well-organized and all information is where it would be expected.
I note some gaps below.
- The authors should state what ages they use to classify children and adolescents. These could range from 2-19. Adolescence could be up to 24.
- Given that the study is looking at wellbeing it would be interesting to know why mental health and wellbeing for children and adolescents is not mentioned in the article.
- It would be interesting to know more about the protocols put in place to safeguard children and adolescents in the study. As noted above, we are not told the ages of the children and adolescents who will be involved. Different protocols will need to be in place for children of different ages to ensure that study participants give informed consent (as noted at line 318).
- It would be interesting to learn more about the ways the authors see this as enabling the city to be redrawn. How will policymakers use this?
Do the authors describe the study protocol adequately?
The title of the article indicates that this study will be redrawing the city. It seems that all the drawing is done in WP1.
The article could say a great deal more about the ways that children and adolescents will be involved in the study.
From the descriptions of the work packages, it appears that children and adolescents will only be involved in WP1. This is certainly an important stage as it will frame the remainder of the work. However, this level of involvement does not justify the claim made in the final paragraph of the discussion section that ‘the main strength of the study is the inclusion of children and adolescents in the development process of the wellbeing framework’. The researchers will refine and develop the conceptual framework over the lifetime of the study. Each of the work packages will lead to adaptations and development. Children and adolescents should be part of this development.
There is a compelling case for involving children and adolescents in each of the work packages so that the findings of the researchers are presented to and debated with the group that is intended to benefit from the study.
WP3 will include children and adolescents but they are described as passive subjects of the study eg ‘educational establishments will be used as points of contact to recruit and collect data on children and adolescents’. Is there a reason for this passive framing?
WP2 and WP4 are technical but there would still be merit in exploring how the study could seek input from children and adolescents at these stages. Dialogue at these stages will help in explaining the technical outcomes of the study to children and adolescents when the project is complete. The researchers will develop a deep understanding of the index but need to learn how to explain their work in ways that hold meaning for different audiences. Constant dialogue will greatly enhance this communication. WP4 recognises the need to make its data available on a web platform – this is a good opportunity to seek input from the children and adolescents who have been involved in the study.
WP5 is presented as a purely technical exercise to validate the wellbeing index. In addition to this necessary technical endeavour it would be straightforward to develop participatory approaches to enable children and adolescents to be part of validating this index. This would enhance its credibility; it would increase ownership of the index and it would increase its political acceptability.
Round 2
Reviewer 1 Report
I see dramatically improvement in the revised version in terms of literature, design, and description. I support that it should be published as such.
Reviewer 3 Report
Thank you for engaging so completely with the comments from the reviewers.
The article now states clearly when and how children and young people will be involved in each work package. You are clear that they will not be involved in the technical work packages eg WP2, WP4, WP5. You are clear that results will be shared with children and young people. This is important. It would be excellent if you were able to revisit WP5 and find ways of involving children and young people in the data validation but I appreciate this may not be possible at this stage.
I look forward to seeing this article published and to learning from the outcomes of the study.
Please check the formatting - some of the paragraphs are single-spaced and others 1.5 spaced.
Please check the resolution of the images - I am sure the editors will pick up on this - they are legible for this review but they are not print quality.
Please check where the citation numbers should go - the square brackets are consistently placed after the full-stop. Should they be before the full-stop?